# Evaluation of the Fitting Accuracy of CAD/CAM-Manufactured Patient-Specific Implants for the Reconstruction of Cranial Defects—A Retrospective Study

**DOI:** 10.3390/jcm11072045

**Published:** 2022-04-06

**Authors:** Henriette L. Moellmann, Vanessa N. Mehr, Nadia Karnatz, Max Wilkat, Erik Riedel, Majeed Rana

**Affiliations:** Department of Oral and Maxillofacial Surgery, University Hospital Duesseldorf, 40225 Duesseldorf, Germany; vanessa.szczepanski@googlemail.com (V.N.M.); nadia.karnatz@med.uni-duesseldorf.de (N.K.); max.wilkat@med.uni-duesseldorf.de (M.W.); mail@erikriedel.de (E.R.); rana@med.uni-duesseldorf.de (M.R.)

**Keywords:** virtual surgical planning (VSP), computer-assisted surgery (CAS), polyetheretherketone (PEEK), patient-specific implants (PSIs), cranioplasty, titanium mesh

## Abstract

Cranioplasties show overall high complication rates of up to 45.3%. Risk factors potentially associated with the occurrence of postoperative complications are frequently discussed in existing research. The present study examines the positioning of 39 patient-specific implants (PSI) made from polyetheretherketone (PEEK) and retrospectively investigates the relationship between the fitting accuracy and incidence of postoperative complications. To analyze the fitting accuracy of the implants pre- and post-operatively, STL files were created and superimposed in a 3D coordinate system, and the deviations were graphically displayed and evaluated along with the postoperative complications. On average, 95.17% (SD = 9.42) of the measurements between planned and surgically achieved implant position were within the defined tolerance range. In cases with lower accordance, an increased occurrence of complications could not be demonstrated. The overall postoperative complication rate was 64.1%. The fitting of the PEEK-PSI was highly satisfactory. There were predominantly minor deviations of the achieved compared to the planned implant positions; however, estimations were within the defined tolerance range. Despite the overall high accuracy of fitting, a considerable complication rate was found. To optimize the surgical outcome, the focus should instead be directed towards the investigation of other risk factors.

## 1. Introduction

Cranioplasty has been indicated to restore the form and function of bone defects of the neurocranium after trauma, intracranial bleeding, or tumor surgery [1,2,3,4,5,6,7,8,9,10,11]. The term “cranioplasty” is defined as a surgical procedure which reconstructs the cranial bone defects of various etiologies [12,13]. It is one of the most common neurosurgical procedures [14]. Craniofacial defects have serious functional and aesthetic consequences, with negative psychological effects [11,15]. In addition to the reconstruction of the shape (symmetry), functional restoration, such as protection, cerebral function, and the improvement of quality of life through surgical measures is essential [11,16]. Reconstruction in the case of sinking scalp flaps or a “syndrome of the trephined” plays a major role, especially in neurological recovery [12,17,18,19,20,21]. Autotransplants are the so-called gold standard in reconstructive surgical therapy. Nevertheless, the use of autologous bone grafts is not free of risk infections; bone resorption or fragmentation may occur. Furthermore, the cranio-maxillofacial area is difficult to reconstruct, especially due to its complex shape, including different curvatures and thicknesses [10,15]. If the autologous bone is not available or suitable, alloplastic materials, such as titanium (Ti), polymethyl methacrylate (PMMA) and ceramics (hydroxyapatites (HA), or polymers, such as polyetheretherketone (PEEK), are used [22,23]. Titanium is the only metal that is used. It is a biocompatible material with a low infection rate [24]. However, titanium has some disadvantages. Its use is associated with high costs and leads to artifacts in imaging [25,26]. Furthermore, the material is very rigid and does not show any protective energy-absorbing properties during traumatic stress [24]. Hydroxyapatite is a ceramic material that is known to be a good scaffold material for bone ingrowth [22]. Unfortunately, due to its brittleness and low tensile strength, its use for larger defects is rather limited [27,28]. PMMA is a polymer that has been widely used due to its low cost, radiolucency, and lack of thermal conductivity. However, there are complications, such as infection, fragmentation and lack of incorporation, which are associated with it [25,29].

Another polymer, PEEK, is also applied in cranial reconstruction [30,31]. PEEK is a stable and highly thermoplastic material. Its sufficient intraoperative fitting and resistance to aggressive sterilization procedures (heat and ionizing radiation) are comparable to the properties of titanium. The elasticity and energy-absorbing properties of PEEK are more similar to those of a bone. The modulus of elasticity of PEEK is 3–4 GPa; in human cortical, bone, it is 7–30 GPa [32,33]. In contrast to titanium, PEEK is a radiolucent, non-magnetic material that facilitates postoperative imaging [24,30,34,35,36]. Nevertheless, PEEK also has minor disadvantages. It has bioactive potential, and the cost of manufacturing a patient-specific PEEK implant is very high [35].

With the help of CAD–CAM technology (Computer-Aided Design/Computer-Aided Manufacturing), implants can be manufactured from all of the above-mentioned material groups [22]. Increasing evidence supports the decision for patient-specific implants (PSIs) for the primary, secondary and subsequent treatment of craniofacial defects [34,35,37]. Based on a CT scan of the patient, the PSI is virtually designed using CAD software and manufactured either by subtraction methods (high-speed milling) or by additive manufacturing techniques [34,35,37]. This widely used approach allows complex reconstructions with precise implant fitting and aesthetic appearance [13,36,38].

The method of fixation of the bone or the implant should be safe, cost-effective, timesaving and aesthetically acceptable. Furthermore, with the minimal use of foreign material, no or minimal artefacts should be seen on neuroradiological imaging [39]. Miniplates are often used to fix the implants, which are particularly convincing due to their biocompatibility and osteointegrative properties [40,41]. With a thickness of only 0.3 mm, excellent cosmetic results can be achieved with good stability [41].

The aim of this retrospective study is to examine the positioning of 39 patient-specific implants (PSIs) made from polyetheretherketone (PEEK) and investigate the relationship between the fitting accuracy and incidence of postoperative complications

## 2. Materials and Methods

We retrospectively evaluated all of the patients who received a PSI for reconstruction (*n* = 39) at the University Hospital Düsseldorf between 2017 and 2020. This study was approved by the local ethics committee (approval number 2018-250). The 39 cases were analyzed regarding demographic, perioperative surgical and medical aspects. To assess the implant fitting, the pre- and post-operative data sets were superimposed, and the postoperative position of the implant was compared with the preoperative digital planning by using heat map analysis. Inclusion criteria were (1) the surgical reconstruction of the cranium using, (2) implants made from PEEK from the manufacturer KLS Martin Group with, (3) virtual 3D planning and computer-aided manufacturing (CAD/CAM process) of the implant, and (4) based on a preoperative CT data set. Exclusion criteria were (1) PEEK implants from other manufacturers, (2) the use of another implant material, and (3) the absence of a postoperative CT scan.

### 2.1. Study Design

This retrospective study considered the analysis of clinical and radiological patient data. On the one hand, the collection and evaluation of demographic and clinical patient parameters were included. On the other hand, the creation and evaluation of heat map analysis determined the accuracy of fitting of the PSI using CT and STL data. The preoperative planning data are available in STL format. These were created by KLS Martin using the IPS Case-Designer program version 1.4 (KLS Martin, Tuttlingen, Germany) and then segmented so that the STL file represented only the planned implant and provided for the present analysis. The programs described below are the 3D editing program Geomagic Freeform Plus (version 2019.1.69), Geomagic Control X (version 2020.0.1.) and iPlan CMF (version 3.0, Brainlab AG, Munich, Germany).

### 2.2. Collection of Demographic and Clinical Data

As part of the data collection, patient- and surgery/implant-related parameters were collected after reviewing the clinical documentation of Düsseldorf University Hospital. These parameters were divided into demographic data, preoperative, intraoperative, and postoperative parameters. Demographic data included name, place of residence, age at the time of implant placement, and gender.

Preoperative parameters considered pre-existing conditions, including treatment history, medication, nicotine use, etiology of the cranial defect and previous cranioplastic reconstructions, including clinical course and explantation information, as well as the time interval between the CT being made in the planning of the implant and implant placement.

Intraoperative parameters are the date of surgery, implant location and size, surgical steps performed in addition to implant insertion (e.g., tumor excision), duration of surgery, any necessary adjustments to the implant or surrounding tissue, and intraoperative complications.

Postoperative parameters describe information on the inpatient stay and postoperative course. These include the clinical course and the duration of the inpatient stay, the time between the operation and the preparation of the postoperative CCT scan, clinical and radiological information on the implant as well as complications during the inpatient stay, clinical and radiological information on the implant during postoperative check-ups, postoperative complications, and information on explantation, if this was performed.

### 2.3. Heatmap Analysis

The creation of the heatmap was conducted following two steps. Initially, the STL file was created, approximating the postoperative implant position (postoperative STL file). Afterwards, this postoperative STL file was compared with the STL file, representing the planned implant position (planning STL file). The following method was used for this purpose:

#### 2.3.1. Creation of the Postoperative STL File

To generate an STL file that represents best the postoperative implant position in situ (i.e., postoperative STL file), the following approach is required:A preoperative CT image in DICOM format, based on which implant planning was performed;A postoperative CT image in DICOM format showing the inserted implant;The planned implant as an STL file (i.e., planning STL file).

The (1) preoperative and (2) postoperative CT images and the (3) planning STL file (red—see Figure 1a,c) were imported into iPlan CMF and automatically fused (see Figure 2). The perforations of the implant and the boundary between the implant and the bone helped with orientation.

#### 2.3.2. Model Comparison

The comparison of the postoperative STL file with the planning STL file in three-dimensional space was performed using the Geomagic Control X. The upper/lower limit for the color coding of the distance measurement values of the postoperative STL file from the reference file was +2 mm and −2 mm, respectively. Deviations from −2 mm to +2 mm are thus displayed in different colors. We used steps of 0.2 mm, so that each color encoded a distance interval of 0.2 mm. All values below −2 mm or above +2 mm are not separately displayed in color but assigned to the lower/upper limits. In this case, all distance values from −2 mm are displayed in dark blue and from +2 mm in red (see Figure 3). The tolerance range for the deviation of the postoperative STL file from the reference file was set to ±1.5 mm. The distances between equivalent points on the STL files were measured. The measured deviations are graphically displayed in a heat map. The evaluation (e.g., minimum/maximum deviation between two equivalent points in mm, proportion of distance measured values within/outside the defined tolerance range in %, and proportion of distance measured values above/below the defined tolerance range in %) are displayed.

### 2.4. Graphical Analysis

The graphical analysis of the heat maps and xBrain files was performed to assess the direction of the deviations. The deviations of the postoperative implant position, in contrast to the planned position, can be described approximately by tilting, rotation and translation. To describe the tilting movement, we examined which pole of the implant (cranial, caudal, rostral, or occipital pole) was tilted proximally (i.e., towards the center of the skull) or distally (i.e., away from the center of the skull). For evaluation purposes, the pole that was most proximal was noted. To describe the rotational movement, we determined (with a viewing direction perpendicular to the implant plane) whether a clockwise or counterclockwise rotation occurred. Since the center of rotation was not centrally located on the implant, which pole of the implant was displaced the most by the rotation and in which direction this occurred were also recorded. To describe the translational movement, the extent to which a displacement of the implant took place in the proximal–distal direction in relation to the cranial center, on the one hand, and in the cranio-caudal and rostro-occipital direction, on the other hand, was examined.

### 2.5. Statistical Analysis

The determined values of the measurements, as well as the clinical data, were statistically analyzed using Jamovi (version 1.6.9, [Computer Software]. Retrieved from https://www.jamovi.org, accessed on 19 March 2022, Sydney, Australia). A *p*-value of <0.05 was defined as significant, a value of <0.01 as highly significant and a value of <0.001 as very highly significant. A significance level of *p* > 0.05 was set for hypothesis testing [42,43]. For associations between two variables, such as A and B, Pearson’s product–moment correlation (r) was calculated if the assumptions of the linear relationship and exclusion of outliers tested via the visual inspection of scatterplots and normal distribution of data, which were assessed using the Shapiro–Wilk test, were met. For non-normal data, Spearman’s rank-order correlation (ρ) was calculated.

## 3. Results

Based on the inclusion criteria, *n* = 39 patient cases in which a cranioplasty procedure was performed using PEEK implants were considered. At the time of implant placement, the patients ranged between 15.3 and 76.3 years of age (M = 44.7; SD = 17.12), with 23 (59%) male and 16 (41%) female subjects. The descriptive data along with the measurements can be found in the Appendix A (Table A1 and Table A2).

### 3.1. Preoperative Parameters

#### 3.1.1. History of Tumor Disease

There was a history of tumor disease in 15 (38.5%, *n* = 15/39) patients of the total collective. Out of these, 5 (12.8%, *n* = 5/39) patients underwent radiotherapy. Among them, radiotherapy was performed before implantation surgery in 4 (80%, *n* = 4/5) of the irradiated patients and after implantation surgery in 1 (20%, *n* = 1/5) patient. In 12 (30.8%, *n* = 12/39) patients, tumor disease was the trigger for subsequent cranioplastic reconstruction; in the other, 3 (7.7%, *n* = 3/39) patients, tumor disease was present outside the cranial region. A history of cardiovascular disease was present in 21 (53.8%, *n* = 21/39) patients. Furthermore, 13 patients were male and 8 were female. Type 2 diabetes mellitus was recognized in 3 patients (7.7%, *n* = 3/39). Thereby, two patients were male, and one patient was female. Obesity was detected in 4 (10.3%, *n* = 4/39) patients, including 1 male and 3 females. Only 1 male (2.6%, *n* = 1/39) patient was underweight. Underweight female patients were not present. Infectious diseases, such as HCV infection, were found anamnestically in 1 (2.6%, *n* = 1/39) male patient.

#### 3.1.2. Nicotine Abuse

Of the 19 patients that made a statement regarding their nicotine use, abuse was present in 9 (47.4%, *n* = 9/19) and no current nicotine use was found in 10 (52.6%, *n* = 10/19) patients. Furthermore, 6 of the 9 patients with positive nicotine use history were male, and 3 were female; 7 of the 10 patients with no current nicotine use were male, and three were female. One patient (10%, *n* = 1/10) from the group of 10 nonsmokers reported former nicotine abuse, who was male.

#### 3.1.3. Etiology of the Cranial Defect

The most frequent indication for craniectomy and thus the etiology of a cranial bone defect in the present patient collective was intracranial neoplasia (30.8%, *n* = 12/39), followed by trauma (25, 6%, *n* = 10/39), ischemic stroke (20.5%, *n* = 8/39), intracranial hemorrhage (17.9%, *n* = 7/39), intracranial abscess (2.6%, *n* = 1/39) and ossification disorder (2.6%, *n* = 1/39). The following intracranial neoplasms occurred: meningioma (41.7%, *n* = 5/12), squamous cell carcinoma (16.7%, *n* = 2/12), glioblastoma (16.7%, *n* = 2/12), astrocytoma (16.7%, *n* = 2/12) and other neoplasms (8.3%, *n* = 1/12). The group of patients with intracranial hemorrhage and aneurysmal subarachnoid hemorrhage (SAB) was slightly more frequent (42.9%, *n* = 3/7) than non-aneurysmal SAB (28.6%, *n* = 2/7). Intracerebral hemorrhage (ICB), isolated or in addition to SAB, was present in all (100%, *n* = 7/7) patients in this group. Thereby, a decompressive craniectomy was performed in 61.5% (*n* = 24/39) of the patients. The following can be noted regarding the gender distribution: Intracranial hemorrhage was observed in four male and three female patients. Half of the patients who had experienced an ischemic stroke were female. Intracranial abscess was present in only one male patient. Ossification disorder, on the other hand, was found only in one female patient.

#### 3.1.4. Previous Cranioplasty

A total of 22 patients (56.4%, *n* = 22/39) had previous cranioplasty. In 40.9% (*n* = 9/22) of the patients in this group, an infection or wound healing disorder was the reason for the explantation of the cranioplastic implant.

#### 3.1.5. Time Interval between Surgery and Planning CT

The mean time interval between obtaining a preoperative CT scan in the planning of the cranioplastic implant and implantation was 61.56 days (M = 61.56; SD = 78.59). The time ranged from 1 day to 419 days, resulting in a range of 418 days. The median was 33 days. The modal value was 112 days.

### 3.2. Intraoperative Parameters

#### 3.2.1. Surgical Duration

The mean duration of implantation surgery was 137.3 min (M = 137.3; SD = 111). The median was 105 min. The modal value was 167 min. The shortest duration of surgery was 38 min. The longest operation time was 602 min, due to microvascular reconstruction. Accordingly, the range was 564 min. Among the patients (*n* = 38), the duration of implantation was not documented for one patient.

#### 3.2.2. Intraoperative Complications

Complications occurred during implant surgery in 20 (51.3%, *n* = 20/39) patients. The most common of these was the perforation of the dura mater (28.2%, *n* = 11/39), followed by the prolapse of the brain with the need for CSF (cerebrospinal fluid) decompression to position the implant in the final position (15.4%, *n* = 6/39), the opening of the frontal sinus (5.1%, *n* = 2/39), the incomplete removal of foreign material during the explantation of the previous cranioplasty (5.1%, *n* = 2/39), flap perforation (2.6%, *n* = 1/39), seizure (2.6%, *n* = 1/39), hemorrhage (2.6%, *n* = 1/39) and periorbital opening during the bony decompression of the orbit (2.6%, *n* = 1/39).

Intraoperative complications occurred more frequently in male patients (60%, *n* = 12/20). Thus, 40% of patients with intraoperative complications were female (*n* = 8/20). Six patients who experienced opening of the dura mater were male. Consequently, the other five patients were female. A prolapse of the brain was present in four male and two female patients. Patients whereby the frontal sinus was opened were all male. Foreign material was left in situ in one female and one male. Flap perforation and seizure during implant surgery occurred in one male patient, respectively. Intraoperative bleeding and opening of the periorbita occurred in one female patient, respectively.

### 3.3. Postoperative Parameters

#### 3.3.1. Length of Inpatient Stay

The average duration of the total inpatient stay was 13.2 days (M = 13.2; SD = 15.1). The median was 8 days. Most frequently, patients stayed on the ward for 4 days (modal value = 4 days). The range of inpatient length of stay was 79 days, with a minimum length of stay of 3 days and a maximum length of 82 days.

If only the duration of inpatient stay after implantation surgery is considered, the average length of stay was 9.72 days (M = 9.72; SD = 13). Half of the patients were in inpatient care for less than 6 days (median= 6 days). The modal value is 3 days. Here, the range was also 79 days, with the shortest duration of inpatient stay being 2 days and the longest duration being 81 days.

#### 3.3.2. Postoperative Complications

Postoperative complications occurred in 25 (64.1%, *n* = 25/39) patients. Of these, 17 (68%, *n* = 17/25) patients were male, and 8 (32%, *n* = 8/25) patients were female. Of these, 14 (35.9%, *n* = 14/39) patients experienced complications during their inpatient stay. A total of 17 (43.6%, *n* = 17/39) patients experienced complications during the postinpatient course. Early complications (<1 month after surgery) occurred only slightly more frequently (43.6%, *n* = 17/39) than late complications (>1 month after surgery) (30.8%, *n* = 12/39). Among these, both early and late complications were present in four patients. Postoperative complications included the following: postoperative bleeding (25.6%, *n* = 10/39), cerebrospinal fluid leakage (20.5%, *n* = 8/39), asymmetry due to muscle atrophy (15.4%, *n* = 6/39), infection (12.8%, *n* = 5/39), worsening of psychomotor status or decrease in vigilance (12.8%, *n* = 5/39), cleft or stunted formation (7.7%, *n* = 3/39), pneumencephalon (7.7%, *n* = 3/39), shunt complications (5.1%, *n* = 2/39), dysesthesias or nerve lesions (5.1%, *n* = 2/39), new-onset neurologic deficit (5.1%, *n* = 2/39), dehiscence (2.6%, *n* = 1/39), subgaleal air collection (2.6%, *n* = 1/39), dislocation of the implant or osteosynthesis material (2.6%, *n* = 1/39) and new-onset seizures (2.6%, *n* = 1/39). In four (10.3%, *n* = 4/39) patients, explantation of the implant was necessary (see Figure 4).

Postoperative complications showed a significant correlation with the occurrence of post-operative bleeding (r = 0.44, *p* = 0.005), CSF leakage (r = 0.38, *p* = 0.017) and asymmetry (r = 0.319, *p* = 0.319). Furthermore, 51.28% of the patients were anticoagulated preoperatively. There was a significant correlation between complications during the inpatient stay and the occurrence of postoperative hemorrhage (r = 0.457, *p* = 0.003), CSF leakage (r = 0.584, *p* < 0.001), shunt complications (r = 0.329, *p* = 0.041), neurological deficits (r = 0.542, *p* = < 0.001) and the occurrence of a pheumencephalon (r = 0.408, *p* = 0.01). In the postoperative course, there was a significant correlation in the occurrence of CSF leaks (r = 0.451, *p* = 0.01) and asymmetry (r = 0.451, *p* = 0.01). Early and late complications were correlated with postoperative hemorrhage (r = 0.465, *p* = 0.003), CSF leak (r = 0.609, *p* < 0.001), neurological deficits (r = 0.46, *p* = 0.003) and pneumencephalon (r = 0.346, *p* = 0.031), respectively, and with infection (r = 0.378, *p* = 0.033) and asymmetry (r = 0.455, *p* = 0.009). The occurrence of infection correlated with the occurrence of dehiscence (r = 0.423, *p* = 0.007) and the need for the explantation of the PEEK implant (r = 0.882, *p* < 0.001) in this sample. If post-operative bleeding occurred, this significantly correlated with the occurrence of neurological deficits (r = 0.653, *p* < 0.001) and pneumencephalon (r = 0.492, *p* = 0.001). Postoperative dehiscence was correlated with the occurrence of CSF leakage (r = 0.319, *p* = 0.048). The presence of a CSF leak was correlated with a subgaleal air collection (r = 0.319, *p* = 0.048), a pneumencephalon (r = 0.33, *p* = 0.04) and neurological deficits (r = 0.375, *p* = 0.019). Implant dislocation was correlated with cleft and step formation (r = 0.562, *p* < 0.001), and dysesthesias (r = 0.698, *p* < 0.001). If asymmetries occurred postoperatively, there was a correlation with new seizures (r = 0.38, *p* = 0.017). If neurological deficits occurred postoperatively, there was a significant correlation with the occurrence of pneumencephalon (r = 0.465, *p* = 0.003).

#### 3.3.3. Follow-Up

A follow-up interval of 147.3 days (SD = 182.79) on average could be realized. However, in half of the patients, the follow-up interval was less than 87 days (median = 87 days). The shortest follow-up interval was 4 days, and the longest is 865 days, resulting in a range of 861 days. The follow-up period comprised the actual time during which the patient was followed up by the University Hospital Düsseldorf, even if an explantation had been already performed. The maximum follow-up interval of 865 days refers to a patient in whom the implant was explanted 12 days after insertion. If only the follow-up interval until explantation is considered, the average is 122.1 days (SD = 135.29).

### 3.4. Fitting Accuracy

#### 3.4.1. Deviation of the Surgically Achieved Implant Position from the Planned Implant Position

On average, 95.17% (SD = 9.42) of the distance measurements (between surgically achieved and planned implant position) were within the defined tolerance range (see Table A2: IO%). The median was 99.72%. The minimum was 56.5%. The maximum corresponded to the modal value and was 100%. This was represented 12 times within the collective. Thus, in 12 (30.77%, *n* = 12/39) of the 39 implants considered, all deviations of the in-situ implant position from the planned position were within the tolerance range of ±1.5 mm. The proportion of distance measurement values that were within the defined tolerance range was 90–100% for 34 implants: 99–99.9% for 11 implants, 95–98.9% for 5 implants and 90–94.9% for 6 implants. For 3 implants, this value was 80–90%; for 0 implants, 70–80%; for 1 implant, 60–70%; for another implant, 50–60%.

The average proportion of distance readings outside the defined tolerance range (NIO%) was consequently 4.83% (SD = 9.42). The smallest proportion was consequently 0%. The implant with the largest deviation from the planned position showed a proportion of out-of-tolerance distance readings of 43.5%. On average, 2.79% (SD = 5.50) of the measured values were above (>+TOL%) and 2.04% (SD = 4.00) of the measured values were below (<−TOL%) the tolerance range. The median of the proportions of distance readings above the defined tolerance range was 0.117%. The median of the proportions of distance measured values below the defined tolerance range was 0.036%. The maximum proportion of values above the tolerance range was 25.48%, and the maximum proportion of values below the tolerance range was 18.02%. The smallest maximum deviation between 2 equivalent points on both implants was 0.35 mm, and the largest maximum deviation was 9.14 mm (in both negative and positive directions). The average maximum deviation in the negative direction (MIN) was 2.28 mm (SD = 1.77). The median was −1.82 mm. The average maximum deviation in the positive direction (MAX) was 2.22 mm (SD = 1.77). The median was +1.82 mm.

#### 3.4.2. Deviations—Graphical

After the graphical analysis of the heatmaps, as well as the xBrain files representing both the planned and the in situ implant, the following can be stated regarding the deviation of the in situ implants: the deviation of 15 of the 39 (38.5%) implants considered can best be described in simplified approximation using a tilting and translational movement; in another 9 (23.1%) implants, by a pure translational movement; in 7 (17.9%) implants, by a tilting and rotation; in 5 (12.8%) implants, by a pure tilting; in 2 (5.1%) implants, by a combination of tilting, rotation and translation; and in one (2.6%) implant, by a pure rotation. The most proximal pole of the implants whose deviation movement can be described (among others) by tilting (29 implants in total) was the cranial pole in 12 implants, the caudal pole in 10 implants, the rostral pole in 4 implants and the occipital pole in 3 implants. When considering the 19 left-sided implants, the most distal pole was the caudal pole in six implants and the cranial pole in six implants. In another three implants, the rostral pole, and in another implant, the occipital pole, were the most distal. Three implants on the left side showed no tilting movement. In the 15 implants on the right side of the skull, the cranial pole was the most distal in five cases; the caudal and occipital poles, in two cases, respectively; and the rostral pole, in one case. Five implants of the right side showed no tilting movement. The most proximally located pole in the five present cranial front implants was the caudal pole in two cases (one implant in each of the right and left front halves) and the cranial pole in one case (left front half). The other two implants did not show any tilting movement. In 10 of the 39 implants considered, the deviation could (among other things) be approximately described by rotation. In total, four implants of the left half of the skull were rotated counterclockwise, and three implants were rotated clockwise. All three implants of the right half of the skull were rotated counterclockwise. The centers of rotation were such that in five cases, the rostral pole was visibly rotated cranially; in three cases, the rostral pole was rotated caudally; the occipital pole was rotated cranially and caudally in one case, respectively. The deviation of 26 implants can be described, among other things, by a displacement in all three spatial planes. An approximately linear movement (along the three possible axes in space) can be assumed for 26 implants. Compared to the planning, seven implants moved proximally (towards the cranial center), and two implants moved distally (away from the cranial center). Six implants were shifted occipitally; five, cranially and occipitally; four, cranially; two, caudally, rostrally, cranially and laterally (left), respectively; one was shifted cranially and rostrally.

## 4. Discussion

### 4.1. Preoperative Parameters

The most frequent indication for craniectomy in the present patient population was intracranial neoplasia (30.8%, *n* = 12/39), followed by trauma (25.6%, *n* = 10/39), ischemic stroke (20.5%, *n* = 8/39), intracranial hemorrhage (17.9%, *n* = 7/39), intracranial abscess (2.6%, *n* = 1/39) and ossification disorders (2.6%, *n* = 1/39). The second most common indication listed here, trauma, is often cited in the literature as the most common reason for decompressive craniectomy, although tumor disease is not listed as an indication [44,45,46]. Broughton et al. (46%), Sobani et al. (45.8%) and Yadla et al. (37.2%) also list trauma as the most common indication, which also includes tumor disease [47,48,49]. In the study by Sobani et al., however, 45.8% only include blunt traumatic brain injury, while penetrating traumatic brain injury is the third most common (11.5%), so the overall rate of traumatic brain injury is even higher [48]. Thus, typical indications for craniectomy in the literature are present in the present analysis. The average surgery duration in the follow-up study of [47] is 93 min. In the present study, the average surgery duration was 137.3 min. The median was 105 min, which shows that an “outlier” leads to a significantly increased average surgery time.

### 4.2. Postoperative Parameters

Postoperative complications after cranioplasty procedures are a serious problem due to the relatively high complication rates [45,50]. The overall complication rate in this retrospective analysis was 64.1% (*n* = 25/39). This is an elevated figure compared to the literature. The patients in this analysis showed 1 (*n* = 9/25) or more (*n* = 16/25) postoperative complications. Additionally, in the study by Brommeland et al., complications not only occur individually, but also multiply [51]. Studies have shown an overall cranioplasty complication rate of 0–28% [52,53,54,55]. The overall complication rate in the present analysis was 64.1% (*n* = 25/39). This includes (1) infection, (2) postoperative hemorrhage, (3) dehiscence, (4) CSF circulation problems (5) subgaleal air collections, (6) implant/osteosynthesis material dislocation, (7) gap/step formation, (8) asymmetry, (9) shunt complications, (10) new-onset seizures, (11) new-onset neurological deficit, (12) dysesthesias/nerve lesions, (13) worsening psychomotor state, (14) pneumencephalon and (15) explantation. The pooled overall complication rate of Malcolm et al. is 19.5% (*n* = 609/3126) and ranges from 3.9% to 45.3% [14]. This also includes infections, rebleeding, dehiscence, CSF circulation disorders, seizures, neurological deficits (but here related to hydrocephalus) and explantation. Subgaleal air collections, dislocations of the implant or osteosynthesis material, cleft and step formation, asymmetry, shunt complications, dysesthesias and nerve lesions, deterioration of the psychomotor state and pneumencephalon are not considered. Bone resorption (in autologous grafts) is additionally considered. Furthermore, studies that included a significant number of non-decompressive craniectomies (e.g., after tumor resection) are excluded [14]. In the present patient population, however, the rate of non-decompressive craniectomies was 35.9% (*n* = 14). The postoperative complication rate in the present patient collective with the indication “tumor disease” (*n* = 12) and, consequently, non-decompressive craniectomy was 75%, which was higher than the overall complication rate, so that with the exclusion of these patients, the complication rate was again slightly lower. Malcolm et al. exclusively refer to an adult patient group without an exact age cut-off. The present collective included 2 patients under 18 years. In their systematic review, Kurland et al. report a lower overall complication rate of 6.4%. Additionally, only new complications after cranioplasty are considered. The complications considered include hemorrhagic and infectious/inflammatory complications, cerebrospinal fluid circulation disorders and bone resorption [56]. However, Kurland et al., in line with Malcolm et al., only considered cranioplasties performed after decompressive craniectomy [14,56]. This differentiates the patient population from this patient population. The overall complication rate reported in the 2015 study by Zanaty et al. was 31.32% (*n* = 109/348). The complications listed therein include postoperative infections, hematomas requiring therapy, hydrocephalus, and seizures. The indications for craniectomy also include bleeding, craniocerebral trauma and strokes. Only infections requiring therapy are also considered. In contrast to the present patient population, death is mentioned as a further complication. Patients with indications for craniectomy tumor intracranial infections and seizures were excluded. However, the present patients included the indications tumor and intracranial abscesses. Furthermore, ossification disorders are not mentioned as an indication. Autologous implant materials (67.24%) are used for the most part. However, according to Zanaty et al., the implant material does not affect the complication rate [45]. The localization of the implants differs from the present localizations; suboccipital cranioplasties are also included. Only hematomas with a need for reoperation are considered, distinct from the present group, in which hematomas without a need for therapy were also considered. The mean age of the patients from the study by Zanaty et al. is 46.5 years (SD = 12.7). The average here was 44.7 years (SD = 17.12). The proportion of male patients is 50.86% (*n* = 177/348), and it was 59% (*n* = 23/39) in this analysis. Diabetes mellitus is present in 14.94% (52/348) in the study and in 7.7% (*n* = 3/39) in the present study. A positive nicotine history was found in 47.13% (*n* = 164/348) and in 52.6% (*n* = 10/19) here. Both former and current smokers are also considered [45]. Thus, the complication rate in the present patient population is higher than the overall complication rates reported in the literature [14,56]; however, it also includes additional complications that are not mentioned in the studies from which the overall complication rates are derived (see above) and includes patient cases with additional indications for craniectomy, such as tumor resection [14,45,56]. The overall complication rate of 64.1% (*n* = 25/39), with the exclusive use of PEEK implants presented here, is significantly higher compared to the overall complication rate with PEEK implants of the systematic review by van de Vijfeijken et al. (21.3%) and also the complication rates of autologous grafts (35.7%) [52,55]. These discrepancies are most likely due to the different patient population regarding, i.e., age and previous diseases. In 87.7% of the studies considered in the review, information was provided regarding age. The mean age is 36 years (from 0 to 90 years) [55]. The collective of this study shows an average age of 44.7 years (SD = 17.12). Wachter et al. show a correlation between higher age ( > 60 years) and postoperative complications [57]. Only 4.8% of the studies considered here report on the proportion of patients with a positive nicotine history [55]. However, this plays a significant role in the development of postoperative complications [57] and amounted to at least 25.6% (*n* = 10/39) in the present collective, so the deviating complication rates may be due to deviating nicotine abuse. Furthermore, only 14% of studies take into account pre-existing conditions that are regularly associated with postoperative complications, resulting in rates of 2.8% for diabetes mellitus, 4.2% for cardiovascular disease, 0.7% for obesity and 6.3% for preoperative radiation [55]. Rates here were 7.7% (*n* = 3/39) for type II diabetes mellitus, 53.8% (*n* = 21/39) for cardiovascular disease, 10.3% (*n* = 4/39) for obesity and 10.3% (*n* = 4/39) for preoperative radiotherapy and other pre-existing conditions, resulting in an overall rate of pre-existing conditions of 76.9% (*n* = 30/39). However, a larger patient population with regard to PEEK implants would be desirable to achieve significance, as PEEK is used as the implant material in only 2.4% of the cranioplasty procedures considered by van de Vijfeijken et al. and is thus the least frequently used material in the review [55]. The systematic review by van de Vijfeijken et al. shows an overall complication rate of 18.9% (*n* = 19,552/10,346) [55]. Autologous cranioplasties show the highest complication rate of 35.7% (*n* = 881), followed by cranioplasties made from titanium, with 22% (*n* = 383); made from PEEK, with 21.3% (*n* = 49); made from PMMA, with 16.8% (*n* = 255); and made from hydroxyapatite, with 10.5% (*n* = 88) [55]. The overall infection rate when considering all of the materials used is 5.6%. Autologous cranioplasties show a significantly higher infection rate of 6.9% than alloplastic materials, with a rate of 5.0%. Hydroxyapatite shows the lowest infection rate (3.3%). The infection rate when using titanium is 5.4%, and for PEEK, it is 5.9%. Polymethyl methacrylate has the highest infection rate at 7.8% [55]. The highest explantation rate (10.4%, *n* = 250) was reported for autologous cranioplasties. For alloplastic materials, the overall explantation rate is 5.1%, with the lowest rate (2.5%, *n* = 21) reported for hydroxyapatite. For PMMA, the explantation rate is 7.9% (*n* = 104); for PEEK, 7.4% (*n* = 18); and for titanium, 6.7% (*n* = 100). The overall explantation rate is 6.6% (*n* = 565) [55].

#### 4.2.1. Infection

The infection rate in the present analysis was 12.8% (*n* = 5/39). Patients in the present collective included in this rate had (1) an infected hematoma in the cranial region with subsequent hematoma evacuation and drainage; (2) a necrotic latissimus dorsi graft in the cranial region with subsequent reoperation; (3) a subgaleal abscess, subdural empyema or epidural abscess with subsequent reoperation; or (4) a wound infection in the cranial region including exposed dura mater with subsequent flap revision and duraplasty. The systematic review by Malcolm et al. from 2016 shows a pooled infection rate of 7.7% (*n* = 165/2021), ranging from 1.4 to 24.4% [14]. Malcolm et al. define the complication “infection” as the totality of all infections that require therapy, such as antibiotic administration, drainage or reoperation. This definition is consistent with the definition used here, so the infection rates are comparable. The combined rate of infectious and inflammatory complications in the systematic review by Kurland et al. from 2015 is 6.0% (*n* = 565/9359) [56]. These include superficial complications, such as wound necrosis, impaired wound healing, surgical wound infections and subgaleal infections, as well as deep complications, such as abscess formation and epidural abscesses and subdural empyema. In addition, meningitis, ventriculitis, resorption of the autologous graft or implant infections, as well as unspecified infections or wound healing disorders at unknown anatomical locations, are considered [56]. The overall infection rate in the 2015 study by Zanaty et al. was 26.43%, of which 14.94% were superficial and 11.49% deep infections. Only infections requiring therapy, which either require reoperation or are treated with antibiotics for at least 2 weeks, were also considered. The patient population in the study by Zanaty et al. shows more postoperative hydrocephalus, a factor associated with the occurrence of postoperative infections [45], which explains the higher rate compared to the analysis presented here. Overall, this makes the present infection rate comparable to the infection rate following cranioplasty procedures reported in the literature [14,45,47].

#### 4.2.2. Postoperative Hemorrhage

Postoperative bleeding occurred in 25.6% (*n* = 10/39) of patients in the present cohort. These include all intracranial and extracranial hemorrhages or hematomas with and without subsequent hematoma clearance, drainage or puncture, as well as marked swelling with suspected hematomas. The pooled rate of hemorrhagic complications in the 2016 systemic review by Malcolm et al. is 4.9% (*n* = 53/1084) and ranges from 2.5% to 7.5%. All intracranial hemorrhages (epidural, subdural and intracerebral) are considered. Extracranial or subgaleal hematomas and swellings with suspected hematoma were not considered. The rate reported by Malcolm et al. again includes extra-axial fluid collections requiring therapy. Kurland et al. report a hemorrhagic complication rate of 3.6% (*n* = 113/3101). Only the main complication mentioned in the studies considered, new ipsilateral hematomas, was considered [56]. The rate of postoperative hematomas requiring reoperation in the study by Zanaty et al. was 6.9% [45]. This includes all intracranial hematomas that occur after implantation and require evacuation. In the present analysis, extracranial hemorrhages and hematomas without therapeutic consequence were also considered. In addition, the present patient population included more male patients. This factor is identified as a predictor for hematomas requiring surgery in the study by Zanaty et al. (OR 2.87, CI 1.13–7.25) [45]. The rate of postoperative bleeding is higher than that reported in the literature. However, the definition of this complication is more broadly defined than in the cited literature.

#### 4.2.3. Dehiscence and CSF Circulation Disorders

The rate of CSF circulation disorders in the present patient population was 2.6% (*n* = 1/39) for dehiscence. This includes wound dehiscence of the skin graft with an exposed PEEK implant. The rate of CSF circulation disorders in the present patient population was 20.5% (*n* = 8/39). These CSF circulation disorders include new or increased extradural CSF leakage or CSF fistulas after implantation and subgaleal CSF accumulation, hydrocephalus, as well as subdural hygromas with and without therapeutic consequences (e.g., placement of a VP shunt or subduroperitoneal shunt, revision of duraplasty, lumbar drainage and punctures). The 2016 systematic review by Malcolm et al. shows a pooled rate of extra-axial fluid collections of 13.9% (*n* = 71/510), ranging from 2.11 to 45.3% [14]. This includes all non-infectious, non-hemorrhagic extra-axial fluid collections, including epidural and subdural fluid collections, hygromas, dural tears and CSF fistulas, so the rates are comparable. The estimated frequency of CSF circulation disorders in the systematic review by Kurland et al. is 5.4% (*n* = 143/2659). These include CSF leakage or CSF fistulas, subdural hygromas and the development of hydrocephalus [56]. In the 2015 study by Zanaty et al., a rate of postoperative hydrocephalus of 13.51% is given. Here, only new cases that occur after cranioplasty and are documented by a CT scan are considered [45]. However, all other CSF circulation disorders mentioned above are not considered, so the rate is lower. In summary, the present rate of CSF circulation disorders is comparable to the rate found in the literature [14,45,56].

#### 4.2.4. Seizures

Seizures occurred postoperatively in 2.6% (*n* = 1/39) of patients. Only new-onset seizures were considered. The 2016 systematic review by Malcolm et al. shows a pooled rate of seizures of 6.1% (*n* = 39/643), ranging from 2.7 to 15% [14]. As in the present study, only new-onset seizures are considered. The 2015 study by Zanaty et al. reported a rate of seizures of 14.37%. In this study, only new-onset seizures were considered. Although the present patient population included more male patients (59%) than the population of Zanaty et al. (50.86%), there was a significantly lower rate of seizures. According to the literature, male gender and age are associated with postoperative seizures [45,58,59,60,61,62]. The rate in the present analysis was overall lower than described in the literature.

#### 4.2.5. General

The complications in the retrospectively followed-up cases presented here are typical complications mentioned in the literature after cranioplasty operations using different materials, so that no atypical or special complications resulted from the use of a PSI made from PEEK. The complication rates did not differ from comparable operations. Thus, it can be assumed that there is no correlation between the use of a PEEK PSI and the occurrence of complications. No correlation could be shown between a suboptimal fitting of the PEEK PSI and the occurrence of complications, as the fitting was very good overall. For the more severe deviations, there was no significant change in the complication rate.

### 4.3. Reasons for the Occurrence of Complications

Since the overall fitting accuracy could be classified as high and, at the same time, there was an above-average complication rate of 64.1%, it can be assumed that other factors are responsible for the complications in addition to the fitting accuracy. Many studies show a limited contribution of surgery-specific factors including the implant material in contrast to patient-specific factors to the overall complication rate [45]. A few selected factors were considered here. The complication rate in patients over 60 years *n* = 9 was 66.7%, and in patients under 60 years, *n* = 30 was 63.3%. Wachter et al. identified older age (>60 years) as a factor associated with postoperative complications [57]. Zanaty et al. show that older age is associated with postoperative seizures and hydrocephalus [45]. The complication rate is 66.7% in patients with pre-existing conditions and 55.6% in patients without pre-existing conditions. Zanaty et al. show that arterial hypertension as a pre-existing condition is associated with a higher postoperative complication rate [45]. In patients who underwent craniectomy due to bleeding, the postoperative complication rate after cranioplasty was 85.7%; in patients with the indications “tumor disease” and “ischemic stroke”, it was 75% in each case. Traumatological patients had a postoperative complication rate of 40%. Patients with an intracranial abscess and ossification disorder showed no postoperative complications. Zanaty et al. were able to show that hemorrhagic stroke predisposes patients to postoperative infections [45]. Common risk factors such as diabetes mellitus and nicotine abuse are mentioned as possible causes [45,63,64]. The postoperative complication rate in patients with a left-sided implant was 66.6% (*n* = 21), whereas patients with a right-sided implant showed postoperative complications in 58.8% (*n* = 17). With only the bilateral implant (bifrontal implant and thus left- and right-sided), the complication rate was 100%. The postoperative complication rate for patients with an implant in the lateral cranium was 58.8% (*n* = 34) and for the localization in the frontal cranium, 100% (*n* = 5). In the study by Gooch et al., the bifrontal location of the implant is significantly associated with postoperative complications [44]. The study by Zanaty et al. distinguishes three localizations of cranioplasty: cranioplasty at the convexity of the skull, and bifrontal and suboccipital cranioplasty. They show that cranioplasty at the convexity has a higher risk of postoperative infections and hematoma. Bifrontal cranioplasties, in turn, are associated with increased rates of seizures and death [45]. Both studies primarily report an association of the bifrontal location with postoperative complications. The present collective with the bifrontal position of the implant showed a complication rate of 100%. However, the result had little significance because the patients mentioned consisted of *n* = 1 person. Suboccipital cranioplasties were not available and, therefore, cannot be compared. The postoperative complication rate in patients who had already experienced intraoperative complications was 65%. Patients who had not experienced intraoperative complications had a rate of 63.2%. Intraoperative injury to the frontal sinus increases the risk of infection after cranioplasty [29,65]. The patients who experienced postoperative complications had a slightly worse implant fitting than the collective without postoperative complications. Although the complication rate increased from 50% to 68% with a decrease in fitting accuracy (from 100 to 90–99%), it then decreased again slightly to 67% with a fitting accuracy of 80–89%. A clear statement about the influence of the fitting accuracy on the complication rate cannot be made for this group of 39 patients. The fitting accuracy was 95.17% (SD = 9.42) on average in the entire collective. In the collective with a below-average time span (<61 days) between planning and surgery, the fitting accuracy was 95.3% (SD = 8.94), and in the collective with an above-average time span (≥62 days), it was 94.8% (SD = 11). This difference is so small that it is not assumed that the time interval between the planning and postoperative stages influenced the fitting accuracy in the present patient collective, provided that the time interval did not exceed the data available here. The implants of the patients in whom the surrounding tissue was processed showed an average accuracy of fit of 94.2% (SD = 11.99); with the processing of the bone margins, 93.5% (SD = 14.33); and with CSF drainage, 97.2% (SD = 4.26). The implants of the patients in whom the surrounding tissue was not processed showed an average fitting accuracy of 95.6% (SD = 8.07); in the case of no bone margin processing, 95.7% (SD = 7.65); and in the case of no CSF drainage, 94.8% (SD = 10.08). Patients who had a CSF drainage showed a slightly higher accuracy of fit. However, since the difference is so small, we do not assume that CSF derivation leads to an unexpectedly higher fitting accuracy. Overall, the findings do not indicate that patients who have had the surrounding tissue processed during surgery will have a poorer fitting.

## 5. Conclusions

The reconstruction of cranial bone defects requires optimal preoperative planning and intraoperative procedures. If the autologous bone is not available or in the case of large or complex defects in the cranio-maxillofacial region, PEEK is an especially promising alternative for reconstruction, along with titanium, HA and PMMA [66]. Due to its mechanical and physical properties, PEEK is often used for cranioplasty. However, a major challenge is to improve the bioactivity, porosity, thickness, biocompatibility, antibacterial ability, integration, and cost reduction in PEEK implants without compromising their mechanical properties. Further research and clinical studies are needed to explore the material properties and possible modifications of cranioplastic applications.

CAD–CAM technology enables the fabrication of customized and patient-specific implants, which showed excellent fit in our study. We were, therefore, unable to establish a correlation between the inadequate fit of the implants and the occurrence of complications. The high postoperative complication rate after cranioplasty is, therefore, mainly due to, for example, patient-specific factors, such as the preoperative conditions, previous diseases, and the duration of the operation. Future studies should, therefore, focus on other risk factors, along with the comparison with other materials, such as titanium.

## Figures and Tables

**Figure 1 jcm-11-02045-f001:**
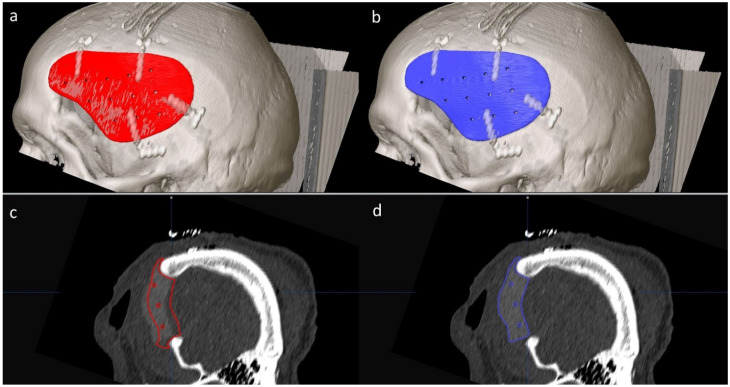
Planned implant position and postoperative outcome in the overview image and a sagittal plane. The red implant shows the implant planning (**a**,**c**). The postoperative implant position is reflected by the blue implant (**b**,**d**). The margins of the planned implant position (red) differed from those of the implant in situ postoperatively (**c**). The postoperative STL file (blue) was positioned so that it was almost congruent with the implant visible in the CT image (**d**).

**Figure 2 jcm-11-02045-f002:**
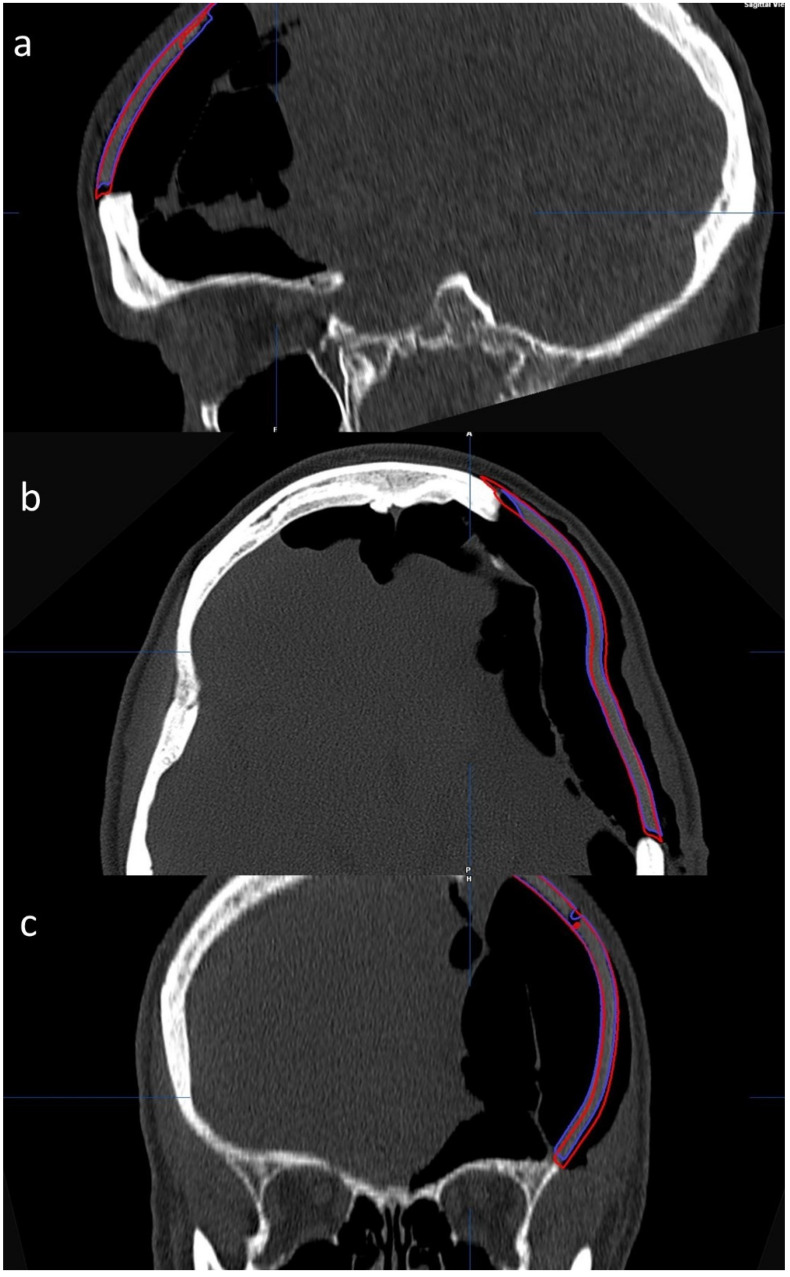
Implant planning and postoperative position in three planes. Positioning of the postoperative STL file (blue) by shifting and rotating was performed in the sagittal (**a**), axial (**b**) and coronal (**c**) planes until it framed the postoperative situation more accurately than the planning STL file (red).

**Figure 3 jcm-11-02045-f003:**
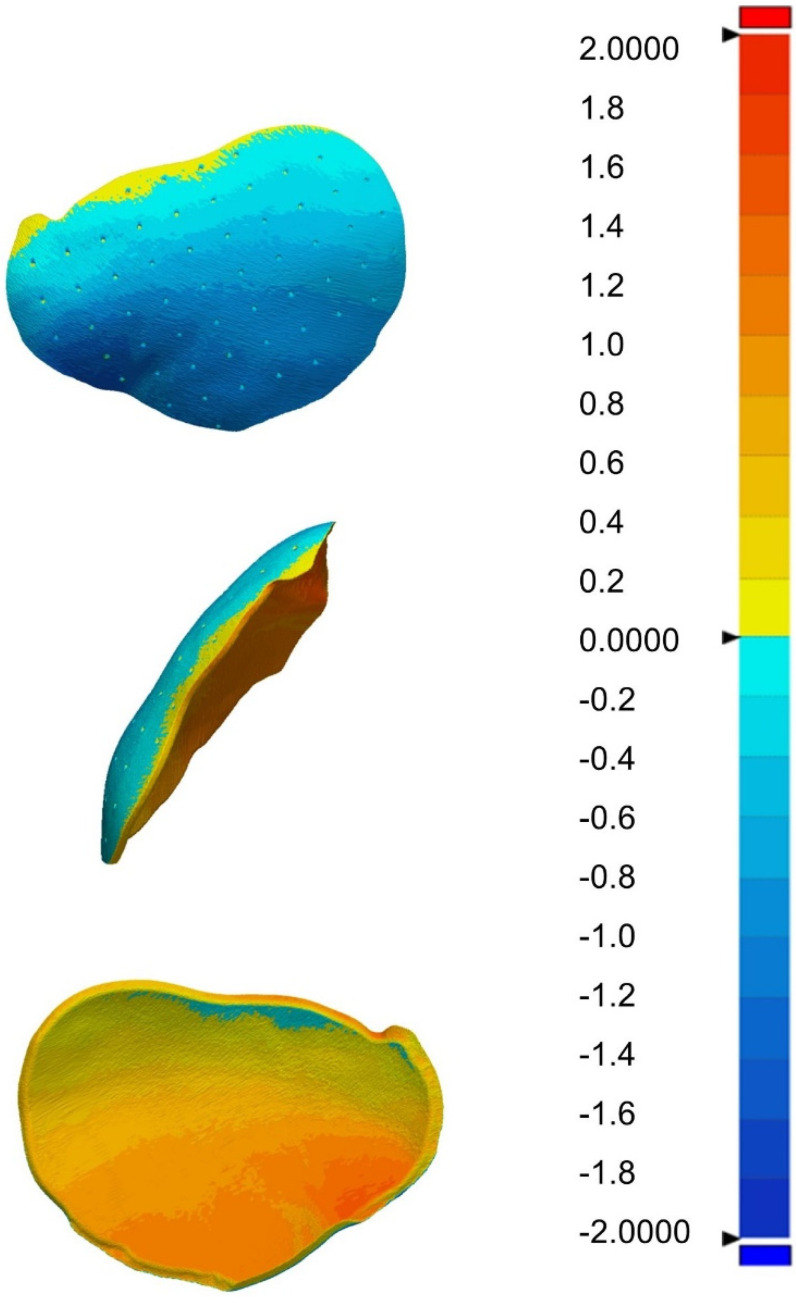
Heatmap of patient no. 19 in three views including the color bar. The heatmap graphically represents the deviation of the postoperative STL file from the planning STL file. Each shade of the color bar encodes a distance interval of 0.2 mm. Values below −2 mm are displayed uniformly in dark blue, and values above +2 mm are displayed uniformly in red.

**Figure 4 jcm-11-02045-f004:**
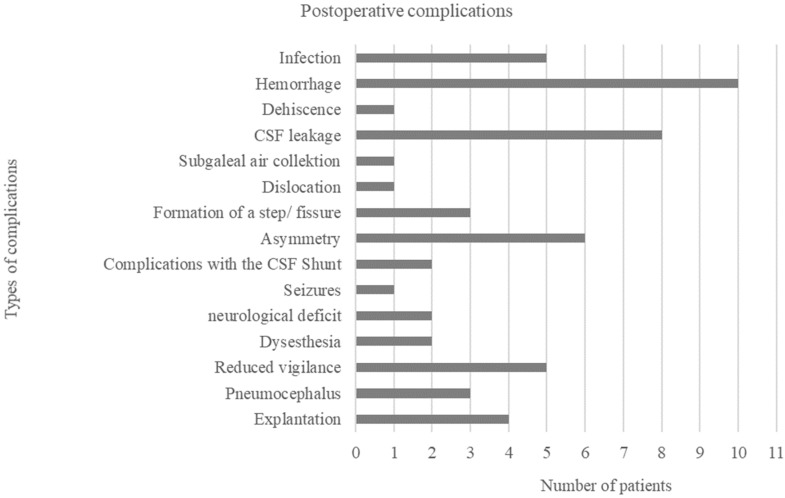
Absolute frequency of postoperative complications.

## Data Availability

The data presented in this study are available on request from the corresponding author. The data are not publicly available due to privacy regulations.

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
