# Peer review of "Evaluation of the Fitting Accuracy of CAD/CAM-Manufactured Patient-Specific Implants for the Reconstruction of Cranial Defects—A Retrospective Study"

_jcm, 2022, doi:10.3390/jcm11072045_

Round 1

Reviewer 1 Report

dear authors    I appreciate your effort in this study and i would like your idea in this study , but i think that there are major errors in this study .  _ scientific writting of manuscript was weak , for example in the abstarct   you did not mention about sample size , statistical anslysis. And it should be revised again. _ keyword should be revised again according to mesh . _ in the introduction , you did not mention aim clearly.  _ there was not enough detail about sample size  and type of study . _ in the statisctical analysis , you should present reference to classification of Pvalue. _ references were old , i think that they should be renewed.   In final, i believe that you should consult with a statistical teacher about sample size.    With regards

Reviewer 2 Report

Dear Authors,

I really appreciate the depth of work and the variables evaluated to understand the origins of such a high complication rate. I agree with the design of the study and with how you demonstrate that the higher incidence of complications could stay in the population. In future, maybe consider to compare another material with PEEK (titanium seems equal from the cited review), trying to keep inalterate the population. The finding would be interesting considering how PEEK vs Titanium behave in a so fragile population. The study is well written, conducted, and developed. 

Correct Line 264  "was"

Reviewer 3 Report

In general the article is interesting and well written and can be accepted for publication after minor revisions:
1) Minor revision of English language because some sentences are too long and are unclear
2) In the materials and methods it would be preferable to explain: a) the surgical procedure as it is performed, the materials used to fix the prosthesis etc...
b) Specify which are the anatomical points of reference used for the colorimetric analysis
3) the results should specify how the various process accuracy groups correspond to the complications rate, because the complications are only listed but not associated to the fitting accuracy groups